REGISTERED REPORT PROTOCOL

# Can post-capture photographic identification as a wildlife marking technique be undermined by observer error? A case study using King Cobras in northeast Thailand

Max Dolton Jones[1]*, Benjamin Michael Marshall[1], Samantha Nicole Smith[1], Jack Taylor Christie[2], Surachit Waengsothorn[2], Taksin Artchawakom[3], Pongthep Suwanwaree[1], Colin Thomas Strine[1]*

1 School of Biology, Suranaree University of Technology, Nakhon Ratchasima, Thailand, 2 Thailand Institute of Science and Technological Research, Nakhon Ratchasima, Thailand, 3 Population and Community Development Association, Bangkok, Thailand

* maxdoltonjones@gmail.com (MDJ); colin_strine@sut.ac.th (CTS)

This is a Registered Report and may have an associated publication; please check the article page on the journal site for any related articles.

## Abstract

Identifying individuals with natural markings is increasing in popularity to non-invasively support population studies. However, applying natural variation among individuals requires careful evaluation among target species, snakes for example have little validation of such methods. Here we introduce a mark-free identification method for King Cobras (*Ophiophagus hannah*) from the Sakaerat Biosphere Reserve, in northeast Thailand using both subcaudal scale pholidosis (scale arrangement and number) and unique ventral body markings to distinguish individuals. This project aims to evaluate the impact of observer error on individual identification. Observers of varying expertise, will distinguish between King Cobra individuals using identifying photographs from a previous study. We will ask randomly assigned observers to distinguish individuals via: 1) subcaudal pholidosis, 2) ventral body markings, and 3) combination of both measures. Using Bayesian logistic regression, we will assess the probability observers correctly distinguish individuals. Based on exploratory observations, we hypothesise that there will be a high probability of correct identifications using subcaudal pholidosis and ventral body markings. We aim to stimulate other studies implementing identification techniques for scrutinous assessment of such methods, in order to avoid subsequent errors during long-term population studies.

## Introduction

Correctly identifying individuals within populations is a fundamental assumption for a number of quantitative population analyses. With confident identification we can estimate abundance [1], density [2], behaviour [3], growth rates [4] and survivorship [5, 6]; thus generating the required information for both management and conservation actions.

Individual marking methods vary and depend upon the focal study species of a study. Non-invasive marking techniques include: neck collars [7], bands/rings [8–10] and external colour marks [11, 12]. Invasive techniques include tags [13], branding [14, 15], transponders [16, 17]

**Data Availability Statement:** All relevant data from this study will be made available via Zenodo upon study completion.

**Funding:** We were supported by the National Science and Technological Development Agency of Thailand via P.S. and Wildlife Reserves Singapore via C.T.S. and M.D.J. The funders had and will not have a role in study design, data collection and analysis, decision to publish, or preparation of the manuscript.

**Competing interests:** The authors have declared that no competing interests exist.

and tissue removal [18, 19]. See Silvy *et al.* [20] for a comprehensive review of available marking methods. A fundamental assumption for many mark-recapture studies is that the mark does not influence survival or behaviour in marked individuals. Invasive marks can indeed be detrimental to long term survival in some species [21].

Despite various animal marking techniques, comparatively few options exist for snakes. Researchers commonly use techniques that cause some degree (though usually minor) of individual bodily harm. Such harm can raise ethical considerations [15, 22, 23]. Methods tailored to snakes include systematic scale clipping [24–27] or branding with cauterising irons [15, 28, 29]. More recently, the ability for researchers to subcutaneously implant passive integrated transponder (PIT) tags efficiently and cheaply, have further increased capacity to identify individuals reliably [17, 30]. Major et al. [31] proposed the novel marking technique of visible implant elastomer (VIE) which can substantially aid in monitoring snake populations long-term, the method is particularly amenable to juvenile and small snake species.

Identification methods using solely morphological or phenotypic traits are increasingly sought after to overcome: the logistical costs of capturing study animals [32], the necessity to identify individuals via camera-trap images [33, 34], and avoiding legislation/ethics restrictions on marking animals [35]. Tiger research studies leverage the natural variation from contrasting black stripes to identify individuals [33, 34]. Other large felines, also have variable characteristics suitable for individual identification, such as bobcats (*Lynx rufus*) [36] and Serengeti cheetahs (*Acinonyx jubatus*) [37]; and alternative methods (variation in tail appearance) have been extremely successful in non-felines such as badgers (*Meles meles*) [38].

Marine studies have relied upon photographic identification tools for decades. Humpback whales (*Megaptera novaeangliae*), manatees (*Trichechus manatus*), and nurse sharks (*Ginglymostoma cirratum*) all demonstrate sufficient variation for individual photographic identification [5, 39, 40]. Meekan et al. [1] used pattern variation and scarring to identify individual whale sharks (*Rhincodon typus*), even a decade after initial sightings.

Attempts to introduce snake focused photographic identification technique however, have been limited. Head patch patterning can be a reliable character for identifying individuals in wild Mangshan pitvipers (*Protobothrops mangshanensis*) [41]. Dyugmedzhiev et al. [42] showed that nose-horned vipers (*Vipera ammodytes*) can be reliably identified using horn scale morphology. While, Bauwens et al. [43] photographed over 3200 individual European adders (*Vipera berus*) and showed that sufficient variation exists even within a large population. Furthermore, Carlström and Edelstam [44] showed that the black and white patterning on the ventral scales of a Swedish population of grass snakes (*Natrix natrix*), could be used to monitor individuals throughout their lifespans, which has led to further support and investigations into *N. natrix* and *N. maura* [45–47].

The King Cobra (*Ophiophagus hannah*), although a large and easily recognisable snake in the field, does not exhibit much variability in phenotypic colour displays. Unless obvious scarring or deformities are present, these snakes are otherwise visually indistinguishable. However, we have now produced an identification method using subcaudal pholidosis and ventral body markings to distinguish between individuals from photographs. Our new method requires only basic photography skills and removes the need for typical invasive identification methods which usually require expertise and permits/licenses to perform.

Although photographic identification methods are increasing in their use, during long-term population monitoring [48], the ability for other researchers to accurately adopt such methods lacks information. Failing to account for observer error when using novel identification techniques can completely undermine the technique altogether leading to incorrect population estimates. Johansson et al. [49] evaluated observer for assessing individual snow leopards (*Panthera uncia*) during photographic surveys with 12.5% erroneous photograph classifications. The

investigation by Johansson et al. [49] suggests that during wildlife camera surveys, or other photographic identification methods, observer error can compromise study findings.

Herein, we propose a study to assess the sensitivity of our novel identification method to observer error. Specifically, we aim to evaluate if observer error is too high that it may undermine our method in being applied to future King Cobra population studies, or potentially extrapolated to other snake species with similar morphological and phenotypical traits.

## Methods

### King Cobra samples

We discovered King Cobras as part of a long-term radiotelemetry study in northeast Thailand using public notifications, active trapping, active surveying and opportunistic captures between 2013 and 2020. Upon capture, we anaesthetised snakes to reduce stress, accurately record biometrics, sex individuals, collect samples and comprehensively photograph study animals. Further information regarding the location, capture and subsequent data collection can be found at Marshall et al. [6, 29, 50]. Photographs from prior capture events will be the main data for this proposed study; however, we only applied basic photography skills to record scalation and body patterning of individuals, and we were not able to standardise lighting conditions and positioning of animals (simulating actual lab conditions in most locations where King Cobra studies might occur).

### Identification

We identified King Cobras captured between 2013 and 2018 (individuals 001–053), using systematic branding following the protocols of Winne et al. [15], and subsequently switched to using PIT-tags (individual 054 onwards); therefore, ensuring correct individual identification facilitating our identification method.

Snake species can have distinguishing subcaudal scale arrangements that is either described as divided or undivided. However, King Cobras in northeast Thailand can possess both divided and undivided scales [51]. Via exploratory analysis, we are confident that the arrangement of divided and undivided scales, and total number of subcaudal scales, is unique to individual King Cobras and can be used for identification, much like a fingerprint.

We preliminarily compared two individual King Cobras and highlighted the observed subcaudal scale arrangement using the software Inkscape (Fig 1), which gives an example of the differences between individuals.

We also investigated whether ventral body markings could further aid identification. King Cobras within the Sakaerat Biosphere Reserve (SBR) typically have yellow/orange colouration on the ventral portion of their hoods, often covered in black or grey peaks on some of the ventral scales. The ventral patterning differs sufficiently to distinguish individuals, highlighted in Fig 2; again, using Inkscape.

We used photographs taken during each King Cobra capture and measurement to compare scalation and body patterning of each known individual. Photographs of each snake's head, body and tail were examined visually, and subcaudal scales counted manually (MDJ). We repeated all counts twice for each individual, adding a third count if the first two counts disagreed (MDJ). From the photographs, we created a unique coding system for each individual based on the transitions from undivided to divided subcaudal scales, similar (though simplified) to a formula suggested by Shine et al. [52]. For example, an individual which has five undivided subcaudal scales, followed by three divided subcaudals, one more row of undivided scales and the remaining 80 scales are divided, would have a code of 5:3:1:80. However, we always start a count with the number of undivided subcaudal scales (the most common arrangement); therefore, if an individual

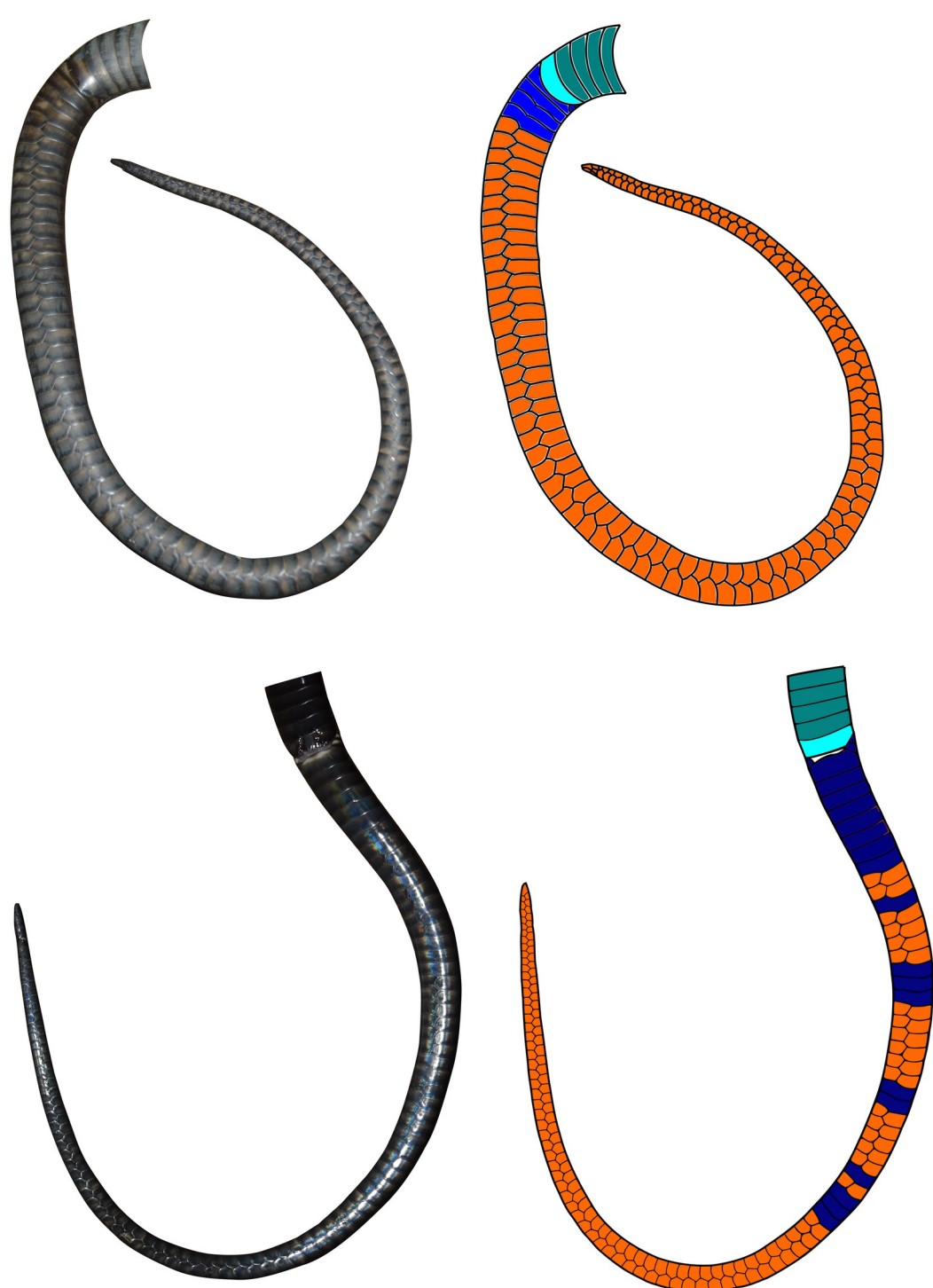

**Fig 1. King Cobra subcaudal patterning.** Subcaudal scale arrangements observed in two King Cobra individuals. Teal: ventral scales, light blue: anal plate, dark blue: undivided subcaudal scales and orange: divided subcaudal scales. Created using Inkscape.

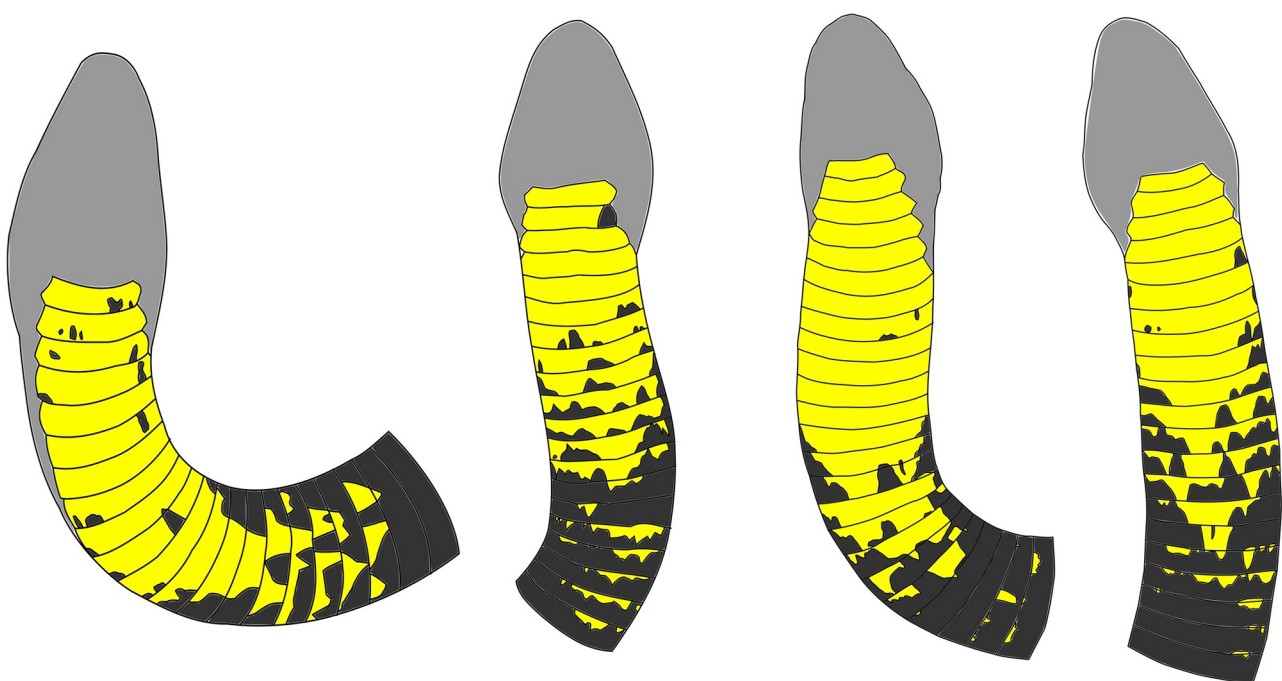

**Fig 2. King Cobra hood patterning.** Ventral hood patterning observed in four King Cobra individuals. Light grey: outline of head, dark grey: dark colouration observed on hood and yellow: yellow/orange colouration observed on hood. Created using Inkscape.

has two divided scales, followed by six undivided and the remaining 90 are divided, the code would be 0:2:6:90.

We currently have a sample size of 81 King Cobras, comprised of 30 adult males, 23 juvenile males, 12 adult females, 4 juvenile females and 12 neonates collected from a single nest. However, research is ongoing and the sample size will likely increase; though we will cease collection and measurement of King Cobras on 01 September 2020. After examining these 81 individuals, we can distinguish all individuals using subcaudal scales alone; and observing ventral body markings when subcaudal counts match (which has occurred on one occasion). We have also compared the subcaudal pholidosis from 16 recaptured individuals (ranging from one to five recaptures) to assess any changes over time. We only observed a change in the total subcaudal scales in one individual (individual 018), resulting from damage to the tip of the tail. Despite this change, we were still able to distinguish this individual due to the observed transition from divided to undivided subcaudal scales. Individual markings can remain the same for at least six years (individual 007), as both subcaudal scale number/arrangement and ventral markings for this individual remained consistent between the two capture events.

As we captured study animals for a radiotelemetry study, we released all individuals back at their capture-site following data collection. We also discovered dead individuals opportunistically across our study site, due to mortality rates [6], which we mostly returned back to discovery site following data collection; however, we stored some individuals within 95% ethanol, currently maintained at the Sakaerat Environmental Research Station, northeast Thailand.

## Assessing identification errors

To validate these measures as a potential identification method, we aim to confirm other observers—besides the authors—can accurately distinguish between individuals. We will create projects on the citizen science platform *Zooniverse* (https://www.zooniverse.org)

requesting the participation of both professional and non-professional citizens to identify individuals within subsets of our available photographs. We will maintain the availability of the project for six months, however, we will increase the duration of the project if a minimum sample of 500 participants, for each sub-project is not reached.

Prior to identifying individuals using our photographs, the site will ask each observer to specify their experience level in snake identification and/or distinguishing individuals within a population. Observers will then self-assign to one of three categories, based on the following criteria: *beginner*, the observer has no experience identifying snakes or individuals within a population; *experienced*, the observer has some applicable skill in identifying snakes/individuals within a population; *expert*, the observer has considerable experience identifying snake species based on morphological characteristics and/or has considerable experience identifying animals to the individual level during population studies.

We will design three projects for volunteers to attempt: 1) Subcaudal scale arrangement, 2) ventral body markings, and 3) a combination of both. Each project will have a tutorial explaining how observers should approach each task, before attempting the actual sets of photos. Each project will contain 50 sets of four images (resulting in 200 photos per project), and each set will contain between one and four King Cobra individuals. Observers will be asked to participate independently and only attempt each set within the project once. Correct answers will only be known to the authors and will not be shared with any participants, even after answers have been submitted. Following the six-month project period, we will share answers using the *Talk* section of the sub-projects, for interested participants.

To remove any influence of the background features of a photograph impacting identification, each snake will be presented on a blank background (Fig 3). Observers will be asked to independently identify how many King Cobras are represented by the four photographs (minimum one, maximum four), also highlighting which photographs represent the same individuals (if any). Observers will also qualitatively rate the average photo quality within the capture event, using the following criteria: *Good*, photos allowed for easy identification of individual subcaudal scales and/or ventral body markings; *Acceptable*, photos allowed for identification of individual subcaudal scales and/or ventral body markings but required careful examination to discern individual scales and/or distinct patterns; *Poor*, photo quality greatly hinder subcaudal scale identification and/or observing ventral body markings and in places did not allow for individual scales to be identified and/or ventral body markings to be observed. A preliminary outline of project workflow can be seen in S1 Fig.

## Analyses

The response variable from observer identification will be binary, and will exhibit a bernoulli distribution. These will classify as follows: 'correct', observer identified the correct number of individuals in an event; or 'incorrect', the observer identified the incorrect number of individuals in an event. We will classify observers' ability to distinguish which individuals are the same using the same binary classification, i.e. 'correct': observers identified which individuals were the same and 'incorrect': observers were not able to identify which individuals were the same.

Using a Bayesian logistic regression model, we will address the two main questions of this study, 1) to what extent can observers identify the correct number of individuals in the capture events? and 2) to what extent can observers identify which individuals within a capture events are the same? We have opted to use a Bayesian logistic regression, as this approach treats model parameters as random variables, which differs to a frequentist approach which considers model parameters as fixed or unknown quantities, and also uses probability to model uncertainty [53]. Furthermore, we can also include prior information derived from literature

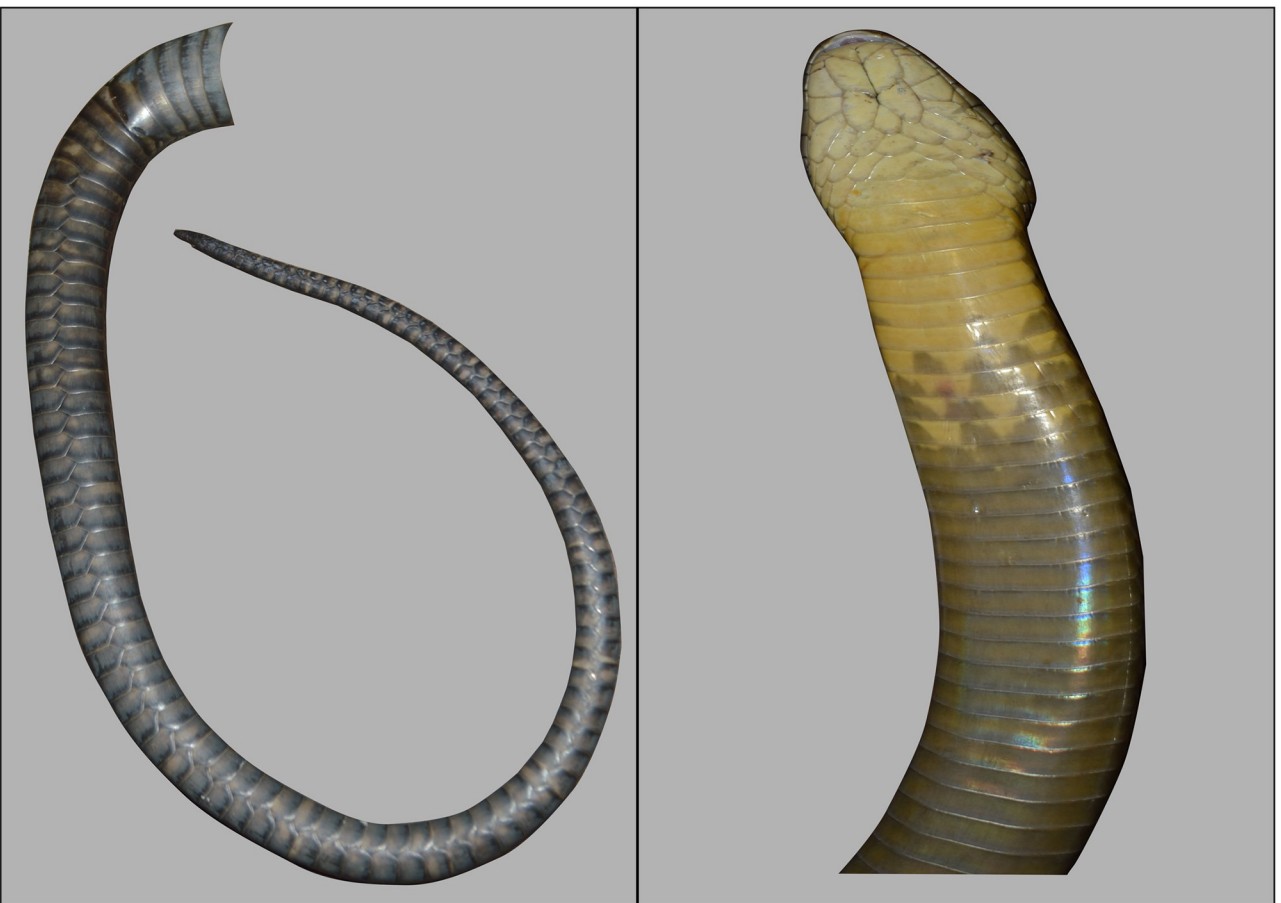

**Fig 3. King Cobra patterning comparison.** Subcaudal scales and ventral body markings of two King Cobra individuals, displayed on a clear background.

review to inform the models. Using Bayes' theorem, a probability distribution will be estimated for our parameters using prior information and observed data [54]. Despite Johansson et al's. [49] study there are few available studies investigating photographic identification observer error (none for snakes), we will therefore be using uninformative priors for the Bayesian regression model (see proposed R code line 55 to 59, S1 File). The first Bayesian logistic regression model will investigate the observer proportion with correct number of individuals in the event. The second model will investigate the proportion of observers correct in showing which individuals were the same in each capture event. We will include three population effects: level of experience, true number of individuals in the capture event, and observer classification of photo quality. We will also include the photo set ID, and observer ID as a group effect.

$$\text{correct} \sim 1 + \text{experience} + \text{true\_number\_indi} + \text{image\_qual} + (1|\text{set\_ID}) + (1|\text{obs\_ID})$$

To address our hypotheses 1) and 2), we will consider any error rate greater than 5% to undermine that particular aspect of the identification method (see proposed R code line 71 to 91 for details, S1 File). In order to validate this identification method for application in future studies, we would need to achieve a probability of observers identifying the number of individuals, and identifying which individuals are the same, of 95% or greater (see proposed R code line 71 to 139 for appropriate outputs, S1 File). Therefore, if either of the Bayesian logistic

models show less than a 95% posterior probability point estimate, we will conclude that observer error does indeed undermine the photographic identification method. This is due to both the need to identify unique individuals within a population (hypothesis 1), and subsequently identify when a unique individual is recaptured (hypothesis 2).

All analyses will be implemented in R v.3.6.3 [55] and R Studio v.1.3 [56]. Bayesian logistic regression models will be performed using the *brms* [57, 58] package. Data will be manipulated and managed using the *scales* [59], *tidybayes* [60], *reshape2* [61], and *dplyr* [62] packages, and subsequent visualisations of data will be created using the *ggplot2* [63] and *ggridges* [64] packages. All package versions will be supplied following analyses.

### Ethics approval and consent to participate

We had ethical approval for the prior data collection from the Suranaree University of Technology Ethics Committee (24/2560), with appropriate Institute of Animals for Scientific Purpose Development (IAD) licences held by C.T.S. Research commenced under the permission by the National Park, Wildlife and Plant Conservation Department, Thailand and the National Research Council of Thailand (98/59). Furthermore, we also had permission to conduct research from Thailand Institute of Scientific and Technological Research and Sakaerat Environmental Research Station. Data supplied by citizens via the Zooniverse.org platform, can be used, modified and redistributed according to the Zooniverse User Agreement and Privacy Policy (Zooniverse.org/privacy).

## Discussion

We aim to assess the further applicability of the King Cobra identification method, by mitigating observer error induced uncertainty, having a baseline prior measure will allow observers to incorporate uncertainty into models.

Ferner and Plummer [65] suggest criteria for selecting suitable marking techniques during study design:

1. "Marks should not affect the survivorship or behaviour of the organism.

2. Marks should allow the animal to be as free from stress and pain as possible.

3. Marks should identify the animal as a particular individual or a member of a cohort if desirable.

4. Marks should last indefinitely or at least through the duration of the study.

5. Marks should be easily read and/or observable by all informed individuals.

6. Marks should be adaptable to organisms of different sizes.

7. Marks should be easy to use in both laboratory and field, and use easily obtained material at minimal cost.

8. Marks should be tested to meet these listed criteria before being put into wide-spread use.

9. Marks should prevent marking application tools from being used without first being thoroughly disinfected and cleaned."

By implementing the identification for King Cobras in the Sakaerat Biosphere Reserve, we can satisfy all of the suggested criteria by Ferner and Plummer [65]. Despite the need for snakes to be captured for subsequent identification, 1) and 2) are achieved through the lack of mutilation or physical tags; however, the need for only a few distinguishing photographs can

substantially decrease handling times, which is specifically important when working with large venomous snakes. We aim to satisfy 3) and 4) by showing that natural marks are unique for each sampled individual in our subset of the population, and recaptures will help fortify this method for long-term population studies. If snakes are required to be captured as part of an ongoing study, 5), 6) and 7) are also achievable as subcaudal scales counts are easy to perform, researchers only need basic skills to photograph appropriate areas of the snake species of interest and standard photography equipment permits the photographing of very small subjects. The purpose of this proposed study is to determine if the methodology is a reliable method for identification and can satisfy the abovementioned criteria (9). Lastly, this methodology allows us to identify individuals without the need for any specialised marking equipment, therefore reducing the need for equipment sterilisation within a field or laboratory setting, outside of standard sterile working environments (10).

The potential to use subcaudal scale anomalies was first suggested by Shine et al. [52], where the authors investigated the subcaudal patterning of 53 blacksnakes (*Pseudechis porphyriacus*) and 115 water pythons (*Liasis fuscus*). Their study showed 41 (77%) *P. porphyriacus* and 56 (49%) *L. fuscus* exhibited unique subcaudal patterning. Shine *et al.* [52] concluded to say that "subcaudal scale formulae may significantly improve the investigator's ability to recognize specific animals on recapture", however, we are yet to find any further studies which have attempted to adopt this methodology.

We hope that the results of this study can be implemented in other photographic identification studies, particularly those focusing on snake population monitoring. This may help to expand the scope of studies which are limited by expertise or licensing to perform the currently accepted marking techniques; namely branding, scale clipping and PIT-tagging. For example, another long-term King Cobra study is being conducted within the Western Ghats, India. In Shankar et al. [35], the authors report on some of the findings of their investigation, specifically involving the capture and biometric data of King Cobras. A quote from their paper states "We could not permanently mark released snakes because we were unable to obtain permission from the Forest Department". This statement alone supports the need for alternative methods of identification for not only King Cobras, but other snake species where permissions and licensing implement road blocks in ongoing research efforts. In the investigation by Shankar et al. [35], snakes were required to be captured and moved as part of a conservation initiative lead by the authors to safely remove unwanted snakes from people's homes. Therefore, since snakes were already being manipulated as part of this service, our proposed methodology could have allowed the researchers to take even the most basic of photographs of the King Cobras for subsequent identification. This could have broadened the scope of their research to investigate if individuals would move back towards removal sites during subsequent re-captures, or even aided towards population estimates and detectability using villager notations; to name a few examples.

We are confident that the proposed methodology can be used as a reliable tool for identifying individuals within our population of interest. However, through the use of external observers, we can evaluate if the wider applicability of our method is undermined by observer error, resulting in misidentifications and subsequent errors in population estimates [49]. We hope to inspire other studies investigating novel identification techniques to assess the resulting observer error before distributing their techniques to the wider scientific community; leading to a more rigorously tested scientific standard for identification methods for wildlife.

If we identify observer error to be too great that it ultimately invalidates our identification strategy, we propose that methodology outlined in Sreekar et al. [66] could be used. Specifically, we could use the Interactive Individual Identification System (I³S Manta), to evaluate if hood patterning can supplementary, or additionally, be used to identify individuals, through a computed measure of similarity, removing the bias of human error from observations.

## Supporting information

**S1 Fig. Zooniverse workflow example.** A preliminary design for the subcaudal scale arrangement project workflow as seen by Zooniverse volunteers.
(TIF)

**S1 File. Simulated Bayesian logistic regression model.**
(R)

## Author Contributions

**Conceptualization:** Max Dolton Jones, Benjamin Michael Marshall, Samantha Nicole Smith, Colin Thomas Strine.

**Data curation:** Max Dolton Jones, Benjamin Michael Marshall.

**Formal analysis:** Max Dolton Jones, Benjamin Michael Marshall.

**Funding acquisition:** Max Dolton Jones, Pongthep Suwanwaree, Colin Thomas Strine.

**Investigation:** Max Dolton Jones, Benjamin Michael Marshall, Samantha Nicole Smith, Jack Taylor Christie, Colin Thomas Strine.

**Methodology:** Max Dolton Jones, Benjamin Michael Marshall, Samantha Nicole Smith, Colin Thomas Strine.

**Project administration:** Max Dolton Jones, Surachit Waengsothorn, Taksin Artchawakom, Colin Thomas Strine.

**Resources:** Surachit Waengsothorn, Taksin Artchawakom.

**Supervision:** Pongthep Suwanwaree, Colin Thomas Strine.

**Validation:** Benjamin Michael Marshall.

**Visualization:** Max Dolton Jones, Benjamin Michael Marshall.

**Writing – original draft:** Max Dolton Jones.

**Writing – review & editing:** Max Dolton Jones, Benjamin Michael Marshall, Samantha Nicole Smith, Colin Thomas Strine.

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
