## [Decision Letter · Decision Letter 0]

23 Oct 2020

PONE-D-20-23862

Can post-capture photographic identification as a wildlife marking technique be undermined by observer error? A case study using king cobras in northeast Thailand.

PLOS ONE

Dear Dr. Strine,

Thank you for submitting your manuscript to PLOS ONE. After careful consideration, we feel that it has merit but does not fully meet PLOS ONE’s publication criteria as it currently stands. Therefore, we invite you to submit a revised version of the manuscript that addresses the points raised during the review process.

We look forward to receiving your revised manuscript.

Kind regards,

Bi-Song Yue, Ph.D

Academic Editor

PLOS ONE

2. In your Methods section, please include a comment about the state of the animals following this research. Were they euthanized or housed for use in further research? If any animals were sacrificed by the authors, please include the method of euthanasia and describe any efforts that were undertaken to reduce animal suffering.

4. Please provide additional details regarding participant consent.

In the ethics statement in the Methods and online submission information, please ensure that you have specified (i) whether consent will be informed and (ii) what type you will obtain (for instance, written or verbal, and if verbal, how it will be documented and witnessed).

If your study included minors, state whether you will obtain consent from parents or guardians.

If the need for consent is waived by the ethics committee, please include this information.

5. Please include captions for your Supporting Information files at the end of your manuscript, and update any in-text citations to match accordingly. Please see our Supporting Information guidelines for more information: http://journals.plos.org/plosone/s/supporting-information

Reviewers' comments:

Reviewer's Responses to Questions

**Comments to the Author**

1. Does the manuscript provide a valid rationale for the proposed study, with clearly identified and justified research questions?

Reviewer #1: Yes

Reviewer #2: Yes

2. Is the protocol technically sound and planned in a manner that will lead to a meaningful outcome and allow testing the stated hypotheses?

Reviewer #1: Yes

Reviewer #2: Yes

3. Is the methodology feasible and described in sufficient detail to allow the work to be replicable?

Reviewer #1: Yes

Reviewer #2: Yes

4. Have the authors described where all data underlying the findings will be made available when the study is complete?

Reviewer #1: Yes

Reviewer #2: Yes

5. Is the manuscript presented in an intelligible fashion and written in standard English?

Reviewer #1: Yes

Reviewer #2: Yes

6. Review Comments to the Author

You may also provide optional suggestions and comments to authors that they might find helpful in planning their study.

Reviewer #1: This Registered Report Protocol addresses an important question regarding the reliability of using unique markings for individual identification, and the progression of methods that maximise the welfare of the study organism. The methodology is feasible, though the authors may wish to consider the use of resources such as zooniverse for attracting participants.

The authors should provide explanation of the acronym SBR in the text (line 140) and there could be improvements to the spelling and language in several places, so I suggest further proofreading prior to final submission.

Where the authors mention their coding system, further information on this would be useful. Furthermore, it would be favourable to reference existing ventral coding systems. Similarly, I would recommend the authors reference other studies using unique ventral patterns for individual snake identification (e.g. Carlstrom and Edelstam (1946): Methods of marking reptiles for identification after recapture; Hailey and Davies (1995) `Fingerprinting' snakes: a digital system applied to a population of Natrix maura; Van Roon et. al. (2006). Capture and recapture of Grass snakes near Amsterdam) as this is absent, despite providing text dedicated to natural markings on other parts of the body and in other taxonomic groups.

The authors could further develop this paper by expanding the variables considered in order to identify characters from the images and animals themselves that may influence identification success.

Reviewer #2: Thanks for the opportunity to review this registered report protocol. The proposed study should be useful for the specific aims of the broader project and also to the wider community of practitioners who could use a similar approach to evaluate the reliability of their own methods. I have only one minor suggestion: you note that that "novel, typically standard, photographic equipment..." is used. But there isn't a lot of detail on how the photographs were actually taken. Were there consistent equipment or lighting protocols or minimum standards for the initial photography? Also, "novel, typically standard" reads as a contradiction, can you clarify?

7. PLOS authors have the option to publish the peer review history of their article (what does this mean?). If published, this will include your full peer review and any attached files.

Reviewer #1: No

Reviewer #2: No

---

## [Author Response · Author response to Decision Letter 0]

7 Nov 2020

Reviewer #1. 

The methodology is feasible, though the authors may wish to consider the use of resources such as Zooniverse for attracting participants.

Line (New) 199 – 237. We thank the reviewer for this suggestion, we had no previous knowledge of such platforms. We created a project on Zooniverse.org to assess its suitability for the project, alongside looking through various other projects and corresponding publications. We decided that this could greatly improve our data collection and have completely re-formatted the “Assessing identification errors” section of the Methods to reflect this new change. We also created a preliminary workflow for the project as it may be seen on the Zooniverse platform, which has now been included as a supplementary figure (S1 Fig). Furthermore, we have now amended our ethics statement to include the use of citizen science data, according to the Zooniverse user agreement.

The authors should provide explanation of the acronym SBR in the text (line 140) and there could be improvements to the spelling and language in several places, so I suggest further proofreading prior to final submission.

Line 140 (New) 150. We have changed “SBR” to “Sakaerat Biosphere Reserve (SBR)”. We have also carefully proofread the manuscript and made comprehensive grammatical changes, alongside new sentence structure, throughout the manuscript.

Where the authors mention their coding system, further information on this would be useful. Furthermore, it would be favourable to reference existing ventral coding systems.

Line 151 (New) 162 – 168. We have cited Shine et al., 1988 as a similar coding system to our own, and have now supplied two examples of our subcaudal coding system to hopefully clarify this to the readers: “…, similar (though simplified) to a formula suggested by Shine et al. (1988). For example, an individual which has five undivided subcaudal scales, followed by three divided subcaudals, one more row of undivided scales and the remaining 80 scales are divded, would have a code of 5:3:1:80. However, we always start a count with the number of undivided subcaudal scales (the most common arrangement), therefore, if an individual has two divided scales, followed by six undivided and the remaining 90 are divided, the code would be 0:2:6:90.”.

Similarly, I would recommend the authors reference other studies using unique ventral patterns for individual snake identification (e.g. Carlstrom and Edelstam (1946): Methods of marking reptiles for identification after recapture; Hailey and Davies (1995) ‘Fingerprinting’ snakes: a digitial system applied to a population of Natrix maural Van Roon et al. (2006). Capture and recapture of Grass snakes near Amsterdam) as this is absent, despite providing text dedicated to natural markings on other parts of the body and in other taxonomic groups. 

Line (New) 99 – 104. We have now included these citations, and Zuiderwijk and Wolterman, 1995, into the introduction within the paragraph exploring natural markings in snakes: “Furthermore, Carlström and Edelstam (1946) showed that the black and white patterning on the ventral scales of a Swedish population of grass snakes (Natrix natrix), could be used to monitor individuals throughout their lifespans, which has led to further support and investigations into N. natrix and N. maura (Hailey and Davies, 1985; Zuiderwijk and Wolterman, 1995; van Roon et al., 2006).”

The authors could further develop this paper by expanding the variables considered in order to identify characters from the images and animals themselves that may influence identification success.

We have found consistent, and reliable, identification using subcaudal pholidosis and ventral hood patterning, and have explored using head scalation to do the same. However, identification using the remaining photos has proven ineffective, and can only bolster identification when prominent scarring or scale blemishes are present. We have therefore opted to focus on the two characteristics which have shown consistent results. Further consideration may cloud inferences as we would need to increase the number of covariates in our models and thereby increase the required sample of participants to infer effects. With Bernoulli trials the corresponding increase in sample size requirements per parameter is considerable. Thus, we prefer to limit the number of features we consider (this is also to provide/refute evidence that this simple and quick method can indeed be a feasible alternative to invasive approaches broadly applied to many non-expert researchers). 

Reviewer #2.

You note that “novel, typically standard, photographic equipment…” is used. But there isn’t a lot of detail on how the photographs were actually taken. Were there consistent equipment or lighting protocols or minimum standards for the initial photography? Also, “novel, typically standard” reads as a contradiction, can you clarify?

Line (New) 311 – 312. We understand the contradiction in our statement, and have changed the sentence to read “…snake species of interest and standard photography equipment permits…”. 

Line (New) 129 – 132. Regarding further explanation on our photography protocols, due to our basic field conditions, we did not have consistent equipment or lighting, and have added the following sentence in the “Sample” heading of the methods: “…; however, we only applied basic photography skills to record scalation and body patterning of individuals, and we were not able to standardise lighting conditions and positioning of animals (simulating actual lab conditions in most locations where King Cobra studies might occur).”.

We hope that we have sufficiently addressed all comments and suggestions, and thank you for your consideration of our registered report protocol for PLOS ONE pending these edits and further review. 

Kind regards,

Colin Strine and Max Dolton Jones

---

## [Editor Report · Decision Letter 1]

10 Nov 2020

Can post-capture photographic identification as a wildlife marking technique be undermined by observer error? A case study using king cobras in northeast Thailand.

PONE-D-20-23862R1

Dear Dr. Strine,

We’re pleased to inform you that your manuscript has been judged scientifically suitable for publication and will be formally accepted for publication once it meets all outstanding technical requirements.

Kind regards,

Bi-Song Yue, Ph.D

Academic Editor

PLOS ONE

---

## [Editor Report · Acceptance letter]

24 Nov 2020

PONE-D-20-23862R1 

Can post-capture photographic identification as a wildlife marking technique be undermined by observer error? A case study using King Cobras in northeast Thailand. 

Dear Dr. Strine:

I'm pleased to inform you that your manuscript has been deemed suitable for publication in PLOS ONE. Congratulations! Your manuscript is now with our production department. 

Kind regards, 

on behalf of

Dr. Bi-Song Yue 

Academic Editor

PLOS ONE